# Taste Interactions between Sweetness of Sucrose and Sourness of Citric and Tartaric Acid among Chinese and Danish Consumers

**DOI:** 10.3390/foods9101425

**Published:** 2020-10-09

**Authors:** Jonas Yde Junge, Anne Sjoerup Bertelsen, Line Ahm Mielby, Yan Zeng, Yuan-Xia Sun, Derek Victor Byrne, Ulla Kidmose

**Affiliations:** 1Food Quality Perception and Society Team, iSense Lab, Department of Food Science, Faculty of Technical Sciences, Aarhus University, 8200 Aarhus N, Denmark; jonas.junge@food.au.dk (J.Y.J.); annesbertelsen@food.au.dk (A.S.B.); lineh.mielby@food.au.dk (L.A.M.); derekv.byrne@food.au.dk (D.V.B.); 2Sino-Danish Center for Education and Research (SDC), 8000 Aarhus, Denmark; zeng_y@tib.cas.cn (Y.Z.); sun_yx@tib.cas.cn (Y.-X.S.); 3Tianjin Institute of Industrial Biotechnology, Chinese Academy of Sciences, Tianjin 300308, China

**Keywords:** cross-culture, individual differences, taste mixtures, model matrix, taste primaries, taste-taste interactions, basic tastes, hierarchical clustering

## Abstract

Tastes interact in almost every consumed food or beverage, yet many aspects of interactions, such as sweet-sour interactions, are not well understood. This study investigated the interaction between sweetness from sucrose and sourness from citric and tartaric acid, respectively. A cross-cultural consumer study was conducted in China (*n* = 120) and Denmark (*n* = 139), respectively. Participants evaluated six aqueous samples with no addition (control), sucrose, citric acid, tartaric acid, or a mixture of sucrose and citric acid or sucrose and tartaric acid. No significant difference was found between citric acid and tartaric acid in the suppression of sweetness intensity ratings of sucrose. Further, sucrose suppressed sourness intensity ratings of citric acid and tartaric acid similarly. Culture did not impact the suppression of sweetness intensity ratings of citric or tartaric acid, whereas it did influence sourness intensity ratings. While the Danish consumers showed similar suppression of sourness by both acids, the Chinese consumers were more susceptible towards the sourness suppression caused by sucrose in the tartaric acid-sucrose mixture compared to the citric acid-sucrose mixture. Agglomerative hierarchical cluster analysis revealed clusters of consumers with significant differences in sweetness intensity ratings and sourness intensity ratings. These results indicate that individual differences in taste perception might affect perception of sweet-sour taste interactions, at least in aqueous solutions.

## 1. Introduction

Taste is an important part of our perception of foods and beverages. Taste perception in humans is considered to consist of the five canonical basic taste qualities: sweet, sour, salt, bitter, and umami [1].

These basic taste qualities interact in almost every consumed food or beverage [2]. Taste interactions occur when at least two tastants are presented simultaneously and affect the perception of each other. Taste interactions can either be enhancing or suppressing, depending on both the taste quality, specific tastants in play, and tastant concentrations [3,4]. Taste interactions are complex, and, even though the interactions between tastes have been extensively researched and reviewed [3,4,5], the mechanisms are still not well understood [6].

This also applies for sweet-sour taste interactions, where both suppression and enhancement have been proposed for both tastes. The general consensus is that sweetness and sourness suppress each other [3,4,7]; however, sweetness enhancement by acids have also been reported [8,9,10]. Only a few studies have investigated other sour stimuli than citric acid in taste interactions with sweetness [11,12]. Moreover, also, very few studies have looked at tartaric acid, but with inconclusive results. Early reports found a sweetness enhancing effect of tartaric acid [8], a claim not supported by later research that has found tartaric acid consistently suppressing sweetness [11,12]. To our knowledge, Fabian and Blum’s [8] is the only study comparing the taste of citric acid and tartaric acid in relation to sweet-sour taste interactions. These authors found a suppression effect of the sourness of both tartaric- and citric acid by sub-threshold levels sucrose. In the same study, they found an unexpected enhancing effect from sub-threshold levels of both tartaric- and citric acid on the sweetness of sucrose.

This inconsistency in findings both related to suppression or enhancement between sourness and sweetness, and to different tastants, could be ascribed to differences in test protocols [3]. Fabian and Blum [8] used the tastant concentrations below taste threshold for the interacting tastant, whereas most others [9,10,11,12,13,14,15,16] use supra-threshold concentrations for both affected and interacting tastant. As Puputti et al. [17] argued, supra-threshold concentrations of both tastants are the most relevant levels in which to investigate taste perception.

A further reason for inconsistencies might be individual differences in perception of taste interactions. Individual differences have not been well studied for taste interactions. McBride and Finlay [18] investigated the effect of sensory training on sweet-sour taste interactions, in a study comparing novices with experienced trained sensory panelists. They found generally high correlations between the two groups for both sweetness suppression by citric acid and sourness suppression by sucrose. Only for the highest concentration of both citric acid and sucrose did they find a difference in sweetness response between the two groups with novices showing higher suppression. The same was not the case for sourness. This indicates low degrees of individual differences in sweet-sour taste interactions. On the other hand, individual differences have been shown to affect taste interactions in other taste combinations. Prescott et al. [7] showed that differences in bitterness sensitivity to 6-*n*-propylthiouracil (PROP) correlated with bitter compounds’ effectiveness in suppressing sweetness, indicating that individual differences might affect taste interactions. In contrast to taste interactions, individual differences in basic taste perception is better understood. Puputti et al. [17] found substantial differences in the perception of basic tastes between individuals. Further, they found evidence suggesting a correlation between supra-threshold taste response to different tastes in one individual, but they suggested, as others before [19], that there is a degree of independent variation in supra-threshold taste response to different tastes in individuals. Studies have found a range of factors that explain some part of supra-threshold taste response, such as genetic differences, age, gender, and weight [20,21,22]. This could potentially also affect taste interactions.

Another group of individual differences that could also influence taste interactions are cultural differences. As for individual differences in general, there are only few studies investigating the effect of culture on taste interactions. Two studies, Prescott et al. [14] and Prescott et al. [15], indicated that even though taste intensities are perceived similarly between Australian and Japanese panelists, some taste interactions might vary with nationality. Again, there is more evidence related to basic tastes than with taste interactions. For example, differences in supra-threshold taste response has been shown in different ethnic groups [23,24]. Williams et al. [23] found both African-American and Hispanic individuals to have a higher supra-threshold taste response than White individuals for all four tastes included in their study. Risso et al. [24] investigated four different vaguely defined ethnic groups, namely Italians, Northern Europeans, Mahgrebis, and Lankans. They investigated the relationship between ethnicity and supra-threshold taste response to mono-sodium glutamate and different bitter compounds. Even though they did find evidence for differences between the ethnic groups in taste sensitivity, no clear pattern in taste sensitivity was seen. Bertino et al. [25] investigated supra-threshold taste response in American and Taiwanese students and found that sweetness of sucrose were influenced by ethnicity both in complex matrices and a simple model matrix, with Taiwanese students ratings sweetness higher than American students.

Conducting cross-cultural comparisons of differences of sweet-sour taste interactions are important as it enables an investigation of whether these established taste interactions, and their magnitudes, are culturally unique or culturally universal [26]. Therefore, the present study had two aims. The first aim was to investigate the effect of different acids, namely citric and tartaric acid, on sweetness- and sourness taste intensity interactions. The second aim was to investigate the perception of taste interactions between sweetness and sourness in a cross-cultural setting. By using sucrose as sweet stimulus and citric and tartaric acid as sour stimuli, all in aqueous solutions, the suppression effect of sweetness on sourness, and the suppression effect of sourness on sweetness were thus investigated using Chinese and Danish consumers. These cultures were chosen as they represent two quite different cultures with very different eating habits [27,28], thus making it more likely that if there indeed are cross cultural differences in taste interactions, they will be present in this context.

When conducting cross-cultural research, a definition of which culture, and which cultures are investigated, is important. Often, cross cultural studies in sensory and consumer science define culture from a geographical point [29,30,31]. As the cross-cultural difference in food perception is thought to be related to eating habits [27,28], being at a certain place at the time of study might not be sufficient to adhere to that food culture. Thus, to further narrow down the cultural differences, nationality is also included in the cultural definition used in this study. Thus, respondents that were both located in Denmark at the time of the study and described their culture as Danish were in this study defined as having a Danish culture. Similarly, respondents that were both in China at the time of study and described themselves as Chinese were considered to have a Chinese culture.

## 2. Materials and Methods 

Taste interaction effects between sucrose, citric acid and tartaric acid were investigated using a 2x3 full factorial design resulting in six samples in total. Relatively low but readily perceivable concentrations of sucrose and acids were chosen in order to make them acceptable for consumers. The concentrations of sucrose and citric acid were chosen based on an internal pilot-study indicating that these concentrations were similar in magnitude of intensity of sweetness and sourness, respectively. The procedure of the internal pilot-study where five internal employees tasted different solutions of citric acid and a sucrose solution with the chosen level of sucrose. After tasting, they indicated which citric acid solution, if any, had similar magnitude as the sucrose solution. To determine equal levels of sourness between citric acid and tartaric acid samples, an iso-sourness study was conducted using a trained sensory panel with eight assessors (7 female, age 42–60), following a protocol adapted after the Sucrose-Sweetener Combined protocol for sweetness by Reference [32], using citric acid as the stereotypical sourness agent. Five different concentrations of both citric acid (0.10, 0.12, 0.14, 0.16, 0.18% *w*/*v*) and tartaric acid (0.80, 0.11, 0.14, 0.17, 0.20% *w*/*v*) were presented. Panelists were provided with three sourness references (being the lowest, medium, and highest concentrations of the citric acid samples) as anchors on the 15 cm line scales used for sourness intensity rating. The anchors were placed at 1.5, 7.5, and 13.5 [7].

A study was conducted afterwards to validate that the sourness of the chosen concentrations of citric and tartaric acid was indeed equal. A trained panel of 10 assessors (9 female, 42–60) evaluated the samples for sourness intensity. No significant differences in sourness intensity were found. Both the iso-sourness study and the validation study were conducted at the sensory facilities at the Department of Food Science, Aarhus University, Aarslev, Denmark.

The six samples were then evaluated by Danish (21st May 2019) and Chinese (24th September 2019) consumers to investigate the effect of culture on the taste interactions.

### 2.1. Samples

Samples were aqueous solutions of either sucrose (Merck KGaA, Darmstadt, Germany), citric acid (Merck KGaA, Darmstadt, Germany) or tartaric acid (L-(+)-Tartaric acid) (Sigma-Aldrich, Co., St. Louis, MO, USA), or combinations of these, as shown in Table 1. Water for the aqueous solutions was San Benedetto, still natural mineral water (San Benedetto S.p.A., Scorzè, Italy). The tastants were weighted out and transferred to 2000 mL volumetric flasks. Tastants were dissolved in approximately 1 L of water. When all tastant was dissolved, the volumetric flask was filled.

After production, 20-mL sample was dispensed into opaque sample tubes with red lids (Fisher Scientific, Roskilde, Denmark), coded with random three-digit numbers. Samples were stored at 5 °C until evaluation, which was at room temperature. The samples were served the day after production.

### 2.2. Consumer Studies

A consumer study was conducted at Aarhus University, Aarhus, Denmark, and at Tianjin Institute of Industrial Biotechnology, Tianjin, China. At both locations, participants were recruited through convenience sampling; thus, both students and staff were recruited to participate in the study and rewarded with a minor gratuity for participation. The consumer characteristics can be found in Table 2. In both places, the consumers conducted the evaluation in a canteen-like area outside eating hours.

Since all ingredients were commercially available food grade products, and the study was conducted in an aqueous model system, the study was exempted from the need of formal ethical approval. However, the panelists did give their verbal consent prior to participation. After consumers agreed to participate in the study, they were asked to complete the questionnaire on their smartphone or on a provided iPad. Consumers received the six samples and were provided with a glass of San Benedetto water.

The participants were given a short verbal introduction to the questionnaire and instructed to check and ensure that the three-digit codes on the tubes matched the ones in the questionnaire, as well as to drink a sip of water between the samples. Consumers were asked to evaluate one sample at a time, and samples were served randomly in accordance with a Williams Latin Square design. All data was collected using the software Compusense (Compusense Inc., Guelph, ON, Canada), which also instructed consumers to cleanse their mouth and wait at least 30 s between the samples.

The questionnaire contained five sample-related questions and nine none-sample related questions. The sample related questions were ratings of sourness and sweetness intensity evaluated on 9-point scales with anchors “Not at all” and “Extremely” [27]. Liking and Just About Right (JAR) for both sourness and sweetness were also evaluated. The questions were presented in the following order: sourness intensity, JAR of sourness, sweetness intensity, JAR of sweetness, and liking. The results from liking and JAR questions are not presented as the focus of this paper is on sweet-sour taste interactions.

Not sample-related questions were culture, gender, age, weight, height, and a 4-question battery of questions using a 7 point Likert scale about general sweet and sour food liking (“I like to eat sweet foods”, “I like to eat sour foods”, “I like to drink fruit juices”, “I like to drink soft drinks”) ranging from “Strongly disagree” corresponding to 1 and “Strongly agree” corresponding to 7. These were all asked after all samples had been evaluated.

Body Mass Index (BMI) was calculated based on consumer height and weight (kg)/(height (m)^2^).

### 2.3. Statistical Analysis

All data analyses were performed using R [33] and RStudio [34]. Data formatting and plotting was conducted using the Tidyverse package [35]. In accordance with the aim only consumers with Danish or Chinese culture were included in the analysis. Further, the age ranged was narrowed to only include consumers in the age 18–30 years of age to ensure a more homogeneous consumer sample. Student’s *t*-tests for comparison of baseline characteristics were conducted using the R package FactoMineR [36].

#### 2.3.1. Overall Sample Evaluation Analysis

To provide an overview of the ratings for sourness intensity and sweetness intensity, means were calculated using R package SensoMineR [37] and Tukey’s Honest Significant Differences (HSD) were used to evaluate pairwise significant differences between samples, considering sample effects as fixed effects and consumer effects as random effects. HSD was determined using the R package lmerTest [38]. Furthermore, Analysis of Variance (ANOVA) was conducted as a mixed model ANOVA using R package SensMixed [39], with Mixed Assessor Model (MAM) to adjust for scaling. Similar to the HSD, in this model, sample effects are considered fixed effects, and consumer effects are considered random effects.

#### 2.3.2. Effect of Acid Quality on Sourness and Sweetness Taste Perception

To investigate the effect of different acids on the sweet-sour taste interaction, interaction plots for the interaction between acid and sucrose for sweetness intensity ratings and sourness intensity ratings, respectively, were compiled. These plots were constructed using R packages lsmeans [40] and ggplot2 [41].

#### 2.3.3. Overall Effect of Culture

Differences in sourness intensity and sweetness intensity, between Danish and Chinese consumers are presented visually using the ggplot2 package in R [42]. They are shown as differences in estimated marginal means between the two groups. Significant differences were calculated using pairwise comparison of estimated marginal means and *p*-values shown are adjusted using Bonferroni corrections [43]. These were calculated using R packages rstatix [44] and emmeans [45].

#### 2.3.4. Effect of Culture on Sweetness- and Sourness Intensity Ratings

To investigate the effect of culture on taste interactions, ratings of sweetness- and sourness intensities were analyzed as follows. Dummy variables were constructed for sucrose and acids indicating whether samples contained this or not. The effects of acid, sucrose, culture and the interactions between these on either sweetness or sourness ratings were analyzed using Mixed model ANOVA with MAM. In the model, the effects of acid, sucrose, and culture were considered as fixed effects, and consumer effects were considered random effects. To further investigate the three-way interactions between acid, sucrose and culture for sourness these were presented visually. This plot was constructed similarly as the interaction plot for the effect of different acids.

#### 2.3.5. Cluster Analysis on Sweetness and Sourness Ratings

Two separate cluster analyses were performed on the ratings of sweetness intensity and sourness intensity, respectively. In both cases, we used Agglomerative Hierarchical Clustering with Euclidean distances and Wards method using the R package cluster [46]. The number of clusters were determined using NbClust package [47] using the majority rule.

Characteristics of consumers in each cluster were summarized, and differences in culture and gender were evaluated using χ2-tests, whereas gender, BMI, sweet food liking, and sour food liking were evaluated with ANOVA. Mean ratings of the different samples were calculated for each cluster, and HSD and ANOVA test for significant differences between clusters for each sample were performed. Mean calculation and HSD were performed similarly to the Overall Sample Evaluation Analysis. ANOVA was performed as one-way ANOVA between samples and clusters for each of the two taste intensity ratings. Lastly, consumers were plotted in PCA, showing both their culture and cluster number using the R package ggfortify [48].

## 3. Results

### 3.1. Sample Evaluation

For an initial overview, all sample means and significant differences in ratings of sweetness and sourness of samples are shown in Table 3. For sweetness ratings, the sample only containing sucrose (S) was highest, followed by the ones containing both sucrose and an acid (SC or ST). This indicates a suppression effect as expected. Interestingly, the sample containing only citric acid (C) was evaluated higher in sweetness ratings than pure water (W), indicating a small sweetness enhancing effect of citric acid. The sample containing only tartaric acid (T) was placed between the two and were not different from neither W nor C.

Regarding the sourness ratings, the expected suppression of sourness by sucrose occurred. The samples containing only acid (C or T) were rated higher in sourness than those containing both sucrose, as well as an acid (SC and ST). As expected, there was no difference between W and S in terms of sourness ratings.

### 3.2. Effect of Acid Quality on Sour and Sweet Taste Interaction

Two different acids were evaluated in this study, namely citric acid and tartaric acid. This was done in order to investigate if there is a difference in the taste interactions between sucrose and different acids. In Figure 1, the results for sweetness and sourness intensity ratings are presented.

Both sourness intensity (Figure 1A) and sweetness intensity (Figure 1B) ratings were not significantly different for the two acids (see Table 3 for significance test). Figure 1A shows that, as expected, citric and tartaric acid differs from the No acid condition, but they are not different from each other. The similarity between the two acids both show that they were rated similarly in sourness intensity but also that citric and tartaric acid had similar sourness suppression by sucrose at the tested concentrations.

In Figure 1B, as expected, the No acid condition differed from the acids when sucrose was present, indicating a sweetness suppression by acids (see Table 3 for significance test). But, also, for sweetness intensity, the acids gave a similar response, indicating a similar suppression effect from both citric and tartaric acid. The citric acid increased sweetness significantly in the No acid condition (Table 3), but Figure 1B shows that the effect of the acids on sweetness intensity ratings are much larger when sucrose is present.

### 3.3. Effect of Culture on Sweet and Sour Taste Interaction

Figure 2 shows differences from the mean in sweetness intensity ratings and sourness intensity ratings (significant differences indicated below) between Danish and Chinese consumers for all samples.

For sweetness intensity ratings, Danish consumers rated the samples consistently higher than the Chinese consumers. This difference is largest for the sucrose containing samples (S, SC, and ST), but still significant for the samples without sucrose (W, C, and T).

There is no clear trend in the differences in the ratings of sourness intensity. Danish consumers rated W and ST highest in sourness intensity, whereas, for T and SC, Chinese consumers rated the highest in sourness intensity. No significant differences were found for C (*p* = 0.15) and S (*p* = 0.40).

#### 3.3.1. Effect of Culture on Sweetness Ratings

The effect of culture on sweet taste ratings was further evaluated by ANOVA. Results can be seen in Table 4.

The Acid × Sucrose × Culture-interaction was not significant (*p* = 0.21), indicating that there was no effect of culture on the suppression of sweetness intensity ratings by acids. This is further underpinned by the insignificance of the Acid × Culture-interaction ((*p* = 0.74), showing that culture of the consumers did not impact the effect of acids on sweetness intensity ratings.

For interactions involving the culture term, only the interaction between culture and sucrose is significant (*p* < 0.01). This means that Chinese and Danish consumers rated the sweetness intensity differently. This was also indicated in Figure 2.

The significant interaction between acid and sucrose for the sweetness intensity ratings (*p* < 0.001) indicates a sweetness suppression effect by acids across the two cultures. This was expected from the results seen in Table 3, where the sweetness intensity of sucrose was suppressed significantly by the acids by a reduction of sweetness intensity ratings.

#### 3.3.2. Effect of Culture on Sourness Ratings

A similar ANOVA was conducted to investigate cultural effects on sour taste ratings (Table 5).

The table shows that the Acid × Sucrose × Culture-interaction term is significant (*p* < 0.01), indicating that there was an effect of culture on the suppression of sourness intensity ratings by sucrose. To investigate this interaction further, the model is presented graphically in Figure 3.

When investigating Figure 3, two differences between the Danish and Chinese consumer groups should be noted. First, even though the sourness intensity ratings of the two acids appear similar when presented alone, citric and tartaric acid behave differently when combined with sucrose in the two consumer groups. Whereas the Chinese consumers had a significantly higher suppression of the sourness ratings of tartaric acid than of citric acid by sucrose, the Danish consumers do not experience any differences, as the confidence intervals overlap. There might be a trend towards the opposite suppression for the Danish consumers compared to the Chinese consumers, namely that the sourness ratings of citric acid might be more suppressed by sucrose than tartaric acid is.

The other thing that comes to mind is that even though the two consumer groups had similar ratings of sourness when sucrose was present, Chinese consumers found the No acid, No sucrose condition less sour than the Only sucrose condition, whereas Danish consumers seemed to find the No acid, No sucrose condition more sour than the Only sucrose condition. Both seem to be trends and, thus, might not be significant.

### 3.4. Cluster Analyses across Culture

The cluster analyses are shown below and reveals patterns of individual difference in ratings of the taste intensities, thus enabling an investigation of differences between different groups of individuals.

#### 3.4.1. Cluster Analysis on Sweetness Intensity

A cluster analysis was performed on sweetness ratings of all consumers to further investigate the differences in sweetness ratings. Three clusters were found, and PCA was performed to further visualize these clusters. PC1 accounts for 35.28% of the variance, and PC2 for 18.22% of the variance, respectively. This can be seen in Figure 4, where both the three clusters are identified, and the culture of the consumers are visualized.

As can be seen in Figure 4, the clusters differentiate rather clearly, even though it should be kept in mind that only slightly more than 50% of the variance is explained by the PCA plot. Further, there might be a visible distinction between Chinese and Danish consumers, where Danish consumers seemed to cluster to the right and Chinese consumers seemed to cluster to the left, though this was not very distinct.

Characteristics of the three clusters are shown in Table 6. The clusters were significantly different for culture (*p* < 0.001) and age (*p* < 0.01). The percentage of female consumers (*p* = 0.35), the BMI (*p* = 0.73), and the liking of sweet (*p* = 0.49) and sour foods (*p* = 0.56) were not significantly different between the different clusters. There was an over-representation of Chinese consumers in Sweet Cluster 1, and an over-representation of Danish consumers in Sweet Cluster 2. It was similar for age, where Sweet Cluster 1 is higher in mean age and Sweet Cluster 2 was lower in mean age. The difference in mean age between those two clusters might be driven by the difference in mean age among the Danish and the Chinese consumer groups.

Regarding the sweetness intensity ratings, the clusters do differ (Table 6). Sweet Cluster 1 generally had low sweetness ratings and a high degree of sweetness suppression effects by acids (difference between S and SC/ST) compared to the other clusters. Sweet Cluster 1 had the lowest sweetness intensity ratings of all clusters for the samples containing both sucrose and acid (ST and SC).

Sweet Cluster 2 differed from the other clusters by on average having higher sweetness ratings for the non-sucrose containing samples. For the samples containing sucrose, the sweetness ratings are higher than Sweet Cluster 1 for both acid-containing samples. For the sucrose and tartaric acid sample (ST), sweetness ratings for Sweet Cluster 2 lies in between Sweet Cluster 1 and Sweet Cluster 3.

Lastly, Sweet Cluster 3 is similar to Sweet Cluster 1 in the average sweetness ratings of the non-sucrose containing samples, and the sucrose only sample. At the same time, Sweet Cluster 3 has higher sweetness intensity ratings of both sucrose and acid-containing samples than Sweet Cluster 1, and, for the sample containing both sucrose and tartaric acid (ST), it has the highest sweetness intensity rating of all clusters. As Sweet Cluster 3 has low sweetness ratings of sucrose-free samples and similarly high ratings of sucrose-containing samples both with and without acids, this cluster shows low suppression effect of sweetness by acids.

#### 3.4.2. Cluster Analysis on Sourness Intensity

Similar to sweetness, a cluster analysis was performed on sourness ratings of all consumers to further investigate the differences in sourness ratings. The PCA shown in Figure 5 depicts the consumers in relation to their sourness ratings. PC1 account for 32.45% of the variance and PC2 for 18.55% of the variance. The consumers do not separate in culture, though there might be a tendency that Chinese consumers are somewhat more to the upper part and Danish consumers in the lower part. On the other hand, the three clusters are very clearly separated. Sour Cluster 3 is a group of more scattered and, in general, fewer consumers than Sour Cluster 1 and Sour Cluster 2. The loadings indicate that Sour Cluster 3 is explaining variance related to the water-only sample. In Table 7, it is seen that this is due to high sourness intensity ratings of this particular sample by Sour Cluster 3.

In Table 7, the characteristics of the three clusters are shown. In baseline characteristics, the three sour clusters are only significantly different in consumer culture (*p* < 0.001). Only Cluster 3 differs by a very high prevalence of Danish consumers. Thus, the age (*p* = 0.70), BMI (*p* = 0.79), and their liking of sweet (*p* = 0.20) and sour foods (*p* = 0.05) are rather similar for all clusters. Further, it should be noted that Sour Cluster 3 is rather small, with only 30 consumers in this cluster.

The sourness intensity ratings certainly differ between the clusters. Sour Cluster 1 and Sour Cluster 2 differ such that Sour Cluster 2 has lower sourness intensity ratings than Sour Cluster 1. This is the case for the two acid-only containing samples (C and T), as well as for the sucrose only containing sample (S) and the sucrose and tartaric acid-containing sample (ST). Interestingly, in Sour Cluster 2 the sourness intensity ratings were not affected by sucrose. Sour Cluster 1 and Sour Cluster 3 only differ in samples W and T. Both Sour Cluster 1 and Sour Cluster 3 have lower sourness intensity ratings of samples containing both sucrose and acids than those just containing acids, indicating a sourness intensity suppression effect from sucrose. Sour Cluster 1 has higher sourness intensity ratings of sample T than Sour Cluster 3. The difference between Sour Cluster 3 and Sour Cluster 1 is mainly due to the oddity of a rather high degree of sourness intensity rating of the water-only sample in Sour Cluster 3.

Interestingly, Sample S is much lower than W in sweetness for Sour Cluster 3, indicating the rated sourness of pure water by Sour Cluster 3 is more or less suppressed by sucrose. It was further investigated whether the high sourness rating of the sample W in Sour Cluster 3 could be due to order effects, but no order pattern were found to be more prevalent in Sour Cluster 3 than in Sour Cluster 1 and 2, indicating that it is not an order effect (data not shown). This was investigated in relation to which number the in W-sample was presented and which sample preceded sample W.

## 4. Discussion

This study investigated taste interactions between sweetness and sourness in Danish and Chinese consumers. Further, it investigated if, and how, these interactions are dependent on sour tastant, thus determining whether citric and tartaric acid give similar sweet and sour responses. An overall significant suppression effect of sweetness from sucrose by both citric and tartaric acid was found. Likewise, overall suppression effect of sourness from both citric and tartaric acid by sucrose was found. Below, the dynamics of these interactions will be discussed in relation to acid quality, as well as both cultural and individual differences.

### 4.1. Effect of Acid Quality on Sweet-Sour Taste Interaction

The effect of citric acid and the effect of tartaric acid were found to be the same both related to sweetness intensity and sourness intensity. Both acids were susceptible to sourness intensity suppression by sucrose to a similar extent, which is consistent with earlier findings [8]. Likewise, both acids suppressed sweetness intensity of sucrose to the same extent. This is in disagreement with the findings of Reference [8], who found sweetness enhancing effects of both acids. This discrepancy might be due to differences in protocols on acid concentration. The concentrations of acids used by Reference [8] were below the taste threshold, thus being well below the concentrations used in this study. Further, studies using a similar concentration of either citric acid [2,49,50] or tartaric acid [12,51] as this study have shown suppression effects similar to those found in this study.

Both sub-threshold and supra-threshold protocols provide valuable information for food perception. Concentrations of tastants as those used in the present study are in the ranges found in beverages, such as fruit drinks [52]. Other beverages, such as soft drinks, might contain considerably higher amounts, which, again, could influence the taste interactions occurring.

### 4.2. Cultural Differences in Taste Interactions

An effect of culture on taste interactions of sweetness on sourness was found but not of sourness on sweetness. Similarities across cultures in taste interactions of sweetness on sourness were also found in an earlier study conducted with Australians and Japanese respondents [15].

Danish consumers, on average, gave higher sweetness intensity ratings than Chinese consumers did. Moreover, an interaction effect was found between sucrose and culture, meaning culture affected the sweetness ratings of sucrose. These effects did not result in effects on taste interactions. This is in agreement with results by Prescott et al. [15], who found a higher sweetness rating by Japanese respondents than Australian respondents but also failed to find taste interactions on sweetness by sourness. Only looking at the taste intensity ratings, and not taste interactions, Bertino et al. [25] similarly found that Taiwanese respondents had a lower sweetness rating than American respondents of aqueous solutions but a higher sweetness response in a complex food matrix.

In contrast to the results found in this study, Prescott et al. [14] did not find cultural differences in taste interactions of sweetness on sourness. The effect found in this study relates to the fact that Chinese consumers displayed a higher sourness suppression by tartaric acid when compared to citric acid, whereas citric and tartaric acid showed the same sourness suppression in Danish consumers. As cultural differences in taste interactions for different acids have not earlier been studied, this is an interesting finding that should be investigated further to see if there are differences for other acids, and if this is similar for other sweeteners.

Besides the difference in susceptibility to sourness suppression of citric and tartaric acid, there was seemingly no consistent effect of culture on sourness ratings.

It is interesting that this study displayed comparable patterns between Danish and Chinese consumers, as Reference [14,15] did between Australians and Japanese respondents. Besides the vague notion of “east versus west”, it is difficult to interpret Australian and Danish respondents as similar culture [31]. The same is evident for Japanese and Chinese respondents. This being said, there have been suggestions that taste sensitivity differs between ethnicity, and Yang et al. [53] found an Asian population to have higher overall taste sensitivity than a Caucasian one.

### 4.3. Individual Differences in Taste Interactions

The cluster analysis for sweetness intensity ratings showed differences between clusters on two parameters, namely sweetness intensity ratings of sucrose-containing samples and sweetness suppression by acids. A possible mechanism could be that these differences could have relation to supra-threshold taste responses. Individual differences in taste response have been reported for all tastes [40,52]. Thus, Sweet Cluster 1 may be said to consist of consumers whom have a lower sweetness sensitivity, as consumers in this cluster respond less to sweetness and experience a higher degree of sweetness suppression from acids. Sweet Cluster 1 includes a larger proportion of Chinese consumers than Danish, indicating that Chinese consumers display a lower supra-threshold taste response. No studies have earlier compared these two groups, but other studies have generally found higher supra-threshold taste response for sweetness in Asian populations [25], and lower in Non-Hispanic White populations [23], indicating the opposite correlation. This could be due to differences in tested concentrations of sucrose, as both Bertino et al. [25] and Williams et al. [23] investigated higher sucrose concentrations than this study.

Further, the low degree of sweetness suppression in Sweet Cluster 3 might be explained by low supra-threshold taste response to sourness, thus affecting lower impact on sweetness. This is more in line with the findings of Williams et al. [23]. Indeed, a relationship between supra-threshold taste response and binary taste interaction have earlier been demonstrated related to PROP sensitivity [7].

Similarly, three factors separate the clusters in the sourness analysis. Here, the main factors are differences in sourness ratings of acids, in suppression of sourness by sweetness, and the surprising sourness of water in Sour Cluster 3. Again, it could be speculated that the differences in sourness is due to differences in taste sensitivity. Sour Cluster 2 show both low sourness response but also low suppression of sourness by sweetness, indicating low sensitivity to both sourness and sweetness. Similarly, Sour Cluster 1 showed high response to sourness and high suppression, which could be an indication of high sensitivity to both sweetness and sourness. This corresponds well to earlier findings that supra-threshold taste response to different tastes for the same individual correlates [54].

The high sourness of water (sample W) in the predominantly Danish Sour Cluster 3 was surprising and difficult to explain. The cluster was somewhat small, so there is a risk that it is just modeling random noise. The effect of order was ruled out by subsequent analysis. Interestingly, the sourness of the water was suppressed by sweetness of sucrose, but Sour Cluster 3 still rated sample S the highest in sourness. As Sour Cluster 3 is only the highest of the clusters in sourness ratings in three of the six samples, it is also unlikely to be an effect of differences in scale use.

This inconsistency in findings both related to suppression or enhancement between sourness and sweetness, as well as to different tastants, could be ascribed to differences in test protocols [3]. Fabian and Blum [8] used the tastant concentrations below taste threshold for the interacting tastant, whereas most others [9,10,11,12,13,14,15,16] use supra-threshold concentrations for both affected and interacting tastant. As Puputti et al. [17] argued, supra-threshold concentrations of both tastants are the most relevant level to investigate taste perception in.

## 5. Conclusions

This study investigated the effect of different acids on sweet-sour taste interactions. It was shown that citric and tartaric acid both showed a similar overall suppression effect of sweetness from sucrose. Likewise, sucrose showed an overall suppression effect of sourness from both citric and tartaric acid. The effects on sourness from citric and tartaric acid were similar.

Further, this study investigated the perception of taste interactions between sweetness and sourness in a cross-cultural setting with Danish and Chinese consumers. An effect of culture on taste interactions of sweetness on sourness was found, but not of sourness on sweetness. Further analysis revealed that sucrose suppressed sourness of tartaric acid to a larger degree in Chinese than Danish consumers.

Lastly, cluster analysis showed differences in taste interactions for both sweetness and sourness between different clusters, with little cultural selection into the groups.

These findings build to the evidence that differences in the perception of taste interactions in beverages are due to the individual rather than cultural differences. Thus, with a focus in the beverage industry to the diversification of the product range, it might be fruitful to investigate not only different demographic and hedonic consumer segments but also perceptual segments.

## Figures and Tables

**Figure 1 foods-09-01425-f001:**
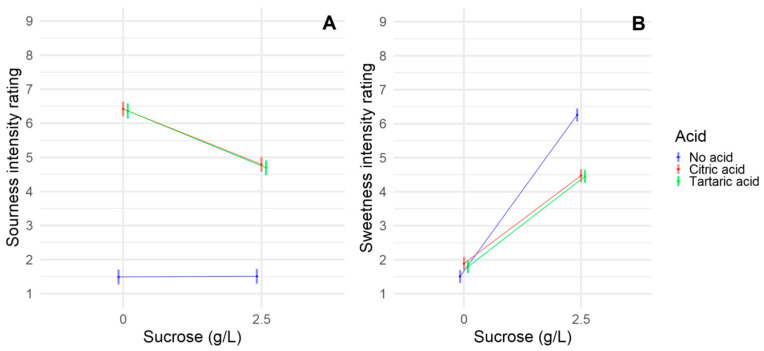
Interaction plots showing the means for the interaction between acids and sucrose. (**A**) Sourness intensity ratings; (**B**) sweetness intensity ratings. Means are full dots. Surrounding faded lines indicate 95% confidence intervals.

**Figure 2 foods-09-01425-f002:**
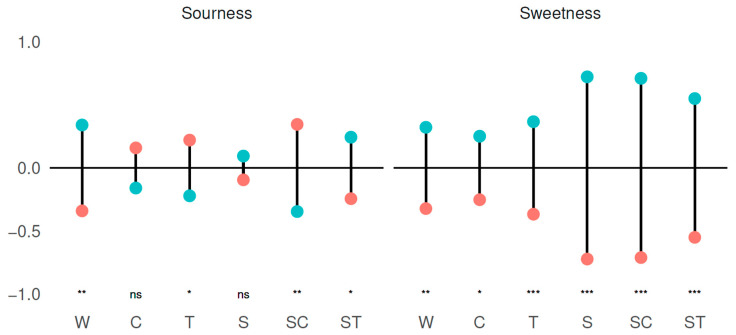
Deviations from means of ratings for sweetness intensity and sourness intensity for Chinese (red dots) and Danish (green dots) consumers, respectively. Samples are Water (W), Citric acid (C), Tartaric acid (T), Sucrose (S), Sucrose and Citric acid (SC), and Sucrose and Tartaric acid (ST). Significant differences are pairwise comparison of estimated marginal means corrected using Bonferroni corrections: ns = not significant, * < 0.05, ** < 0.01, *** < 0.001.

**Figure 3 foods-09-01425-f003:**
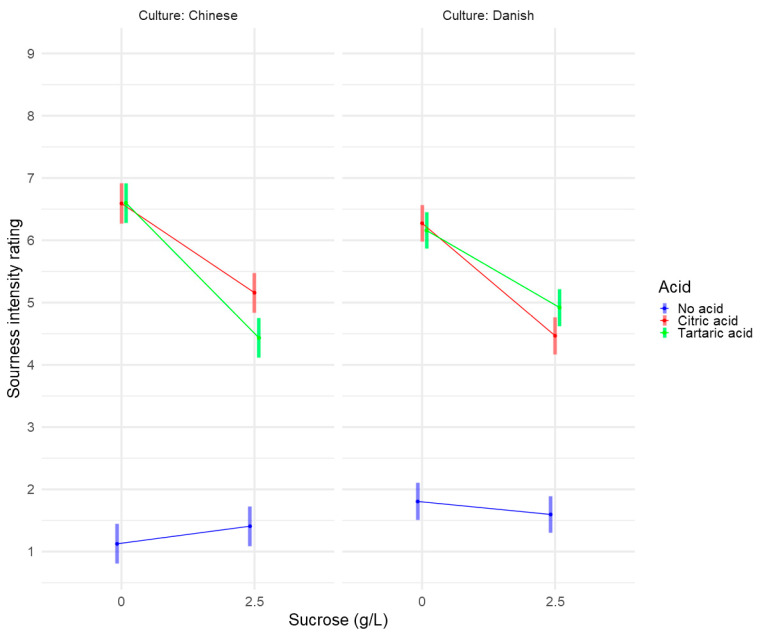
Interaction plot showing the means of the interaction between acids and sucrose for sourness intensity rating for Chinese and Danish consumers, respectively. Means are full dots. Surrounding faded lines indicate 95% confidence intervals.

**Figure 4 foods-09-01425-f004:**
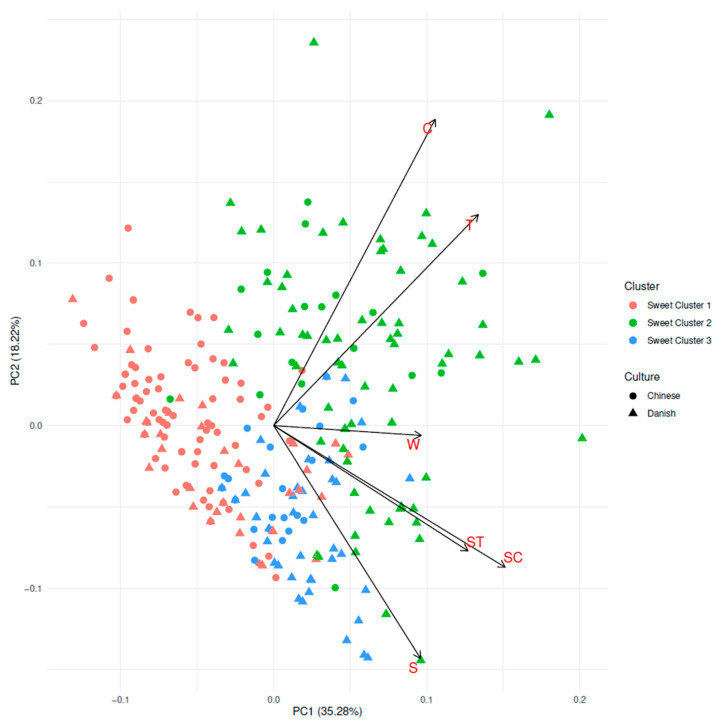
PCA biplot showing consumers sweetness ratings. Both their relation to clusters and culture are visualized. Samples are Water (W), Citric acid (C), Tartaric acid (T), Sucrose (S), Sucrose and Citric acid (SC), and Sucrose and Tartaric acid (ST).

**Figure 5 foods-09-01425-f005:**
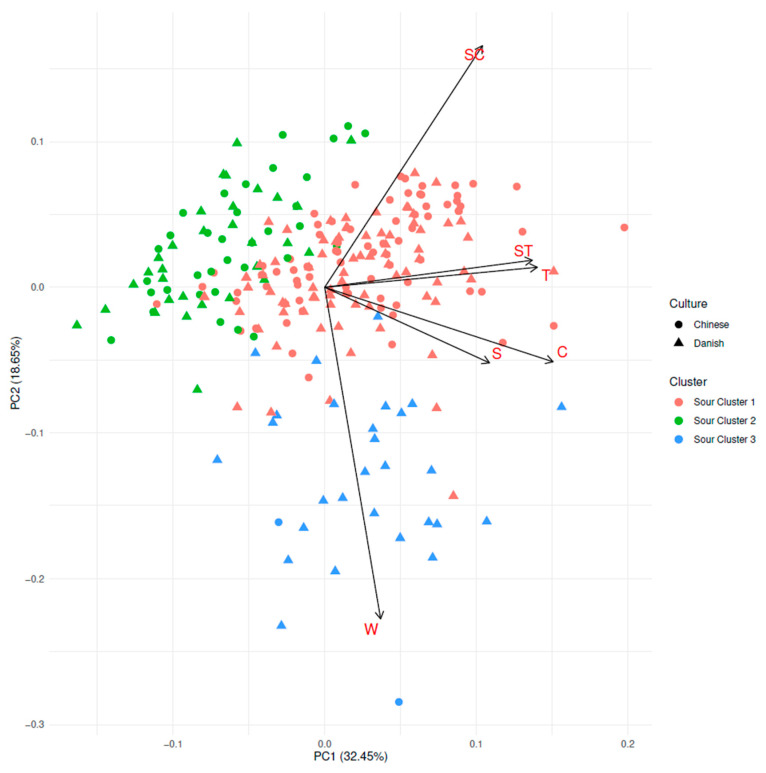
PCA biplot showing consumers sourness ratings. Both their relation to clusters and culture are visualized. Samples are Water (W), Citric acid (C), Tartaric acid (T), Sucrose (S), Sucrose and Citric acid (SC), and Sucrose and Tartaric acid (ST).

**Table 1 foods-09-01425-t001:** Sample names and concentrations of tastants. Samples are Water (W), Citric acid (C), Tartaric acid (T), Sucrose (S), Sucrose and Citric acid (SC), and Sucrose and Tartaric acid (ST).

Sample	Sucrose (% *w*/*v*)	Citric Acid (% *w*/*v*)	Tartaric Acid (% *w*/*v*)
W	-	-	-
C	-	0.140	-
T	-	-	0.114
S	2.50	-	-
SC	2.50	0.140	-
ST	2.50	-	0.114

**Table 2 foods-09-01425-t002:** Baseline characteristics of Danish and Chinese consumers. Provided *p*-values of Student’s *t*-test for difference between Danish and Chinese consumer groups. The age range is presented in parenthesis. Sweet Foods Liking and Sour Foods Liking rated in 7-point Likert scale (7 = most liked).

	Denmark (*n* = 139)	China (*n* = 120)	*p*-Value
Mean Age (years)	23.24 (19–30)	25.34 (21–30)	<0.01
Number of Female	88 (62.9%)	79 (65.3%)	0.59
Mean BMI ^1^	23.37	21.47	<0.01
Sweet Foods Liking	5.28	5.06	0.23
Sour Foods Liking	5.03	4.25	<0.01

^1^ BMI is Body Mass Index.

**Table 3 foods-09-01425-t003:** Means of sweetness and sourness ratings for all samples, as well as F- and *p*-values for Analysis of Variance model. Samples are Water (W), Citric acid (C), Tartaric acid (T), Sucrose (S), Sucrose and Citric acid (SC), and Sucrose and Tartaric acid (ST). Letters are to be read row-wise, and they indicate significant differences between samples based on Honest Significant Differences (HSD).

	W	C	T	S	SC	ST	F-Value	*p*-Value
Sweetness	1.53a	1.90b	1.82ab	6.32d	4.53c	4.49c	505.34	<0.001
Sourness	1.49a	6.42c	6.36c	1.51a	4.79b	4.70b	496.78	<0.001

**Table 4 foods-09-01425-t004:** F- and *p*-values for mixed model Analysis of Variance for Sweetness Intensity Ratings. Significant effects (*p* ≤ 0.05) are marked in bold.

	F-Value	*p*-Value
Acid	53.98	**<0.001**
Sucrose	848.55	**<0.001**
Culture	70.87	**<0.001**
Acid × Sucrose	112.98	**<0.001**
Acid × Culture	0.31	0.74
Sucrose × Culture	9.01	**<0.01**
Acid × Sucrose × Culture	1.55	0.21

**Table 5 foods-09-01425-t005:** F- and *p*-values for mixed model Analysis of Variance for Sourness Intensity. Significant effects (*p* ≤ 0.05) are marked in bold.

	F-Value	*P*-Value
Acid	1274.37	**<0.001**
Sucrose	131.60	**<0.001**
Culture	0.02	0.90
Acid × Sucrose	53.72	**<0.001**
Acid × Culture	12.74	**<0.001**
Sucrose × Culture	0.01	0.91
Acid × Sucrose × Culture	8.93	**<0.01**

**Table 6 foods-09-01425-t006:** Cluster characteristics and means of ratings for Sweetness in Sweet clusters. Samples are Water (W), Citric acid (C), Tartaric acid (T), Sucrose (S), Sucrose and Citric acid (SC), and Sucrose and Tartaric acid (ST). *P*-values for cluster culture and gender are for χ2 analysis of difference, whereas *p*-values for Age, BMI, and Sweet and Sour Food Liking are from ANOVA. *P*-values for Sweetness Intensity are derived from mixed model Analysis of Variance. Letters are to be read row wise, and indicate significant differences between samples based on Honest Significant Differences (HSD). Note that, even though the Analysis of Variance find significant differences between clusters for sample S, HSD fail to identify differences. Significant effects (*p* ≤ 0.05) are marked in bold.

	Sweet Cluster 1	Sweet Cluster 2	Sweet Cluster 3	*p*-Value
Total Consumers in cluster	110	85	64	-
Mean of Age	24.9	23.6	23.9	**<0.01**
Number of Chinese (%)	78 (70.9)	18 (21.2)	24 (37.5)	**<0.001**
Number of Female (%)	76 (69.1)	52 (61.2)	38 (59.4)	0.35
BMI	22.3	22.8	22.4	0.73
Sweet Food Liking	5.2	5.2	5.3	0.49
Sour Food Liking	4.7	4.8	4.5	0.56
W	1.08a	2.35b	1.20a	**<0.001**
C	1.26a	3.04b	1.50a	**<0.001**
T	1.26a	2.88b	1.36a	**<0.001**
S	5.80a	6.75a	6.62a	**<0.01**
SC	3.25a	5.27b	5.73b	**<0.001**
ST	3.09a	4.95b	6.28c	**<0.001**

**Table 7 foods-09-01425-t007:** Cluster characteristics and means of ratings for Sourness Intensity in clusters. Samples are Water (W), Citric acid (C), Tartaric acid (T), Sucrose (S), Sucrose and Citric acid (SC), and Sucrose and Tartaric acid (ST). *P*-values for cluster culture and gender are for χ2 analysis of difference, whereas *p*-values for Age, BMI, and Sweet and Sour Food Liking are from ANOVA. *P*-values for Sourness Intensity are from mixed model Analysis of Variance. Letters are to be read row wise, and indicate significant differences between samples based on Honest Significant Differences (HSD). Significant effects (*p* ≤ 0.05) are marked in bold.

	Sour Cluster 1	Sour Cluster 2	Sour Cluster 3	*P*-Value
Total Consumers in cluster	164	65	30	-
Mean of Age	24.4	24.1	23.6	0.70
Number of Chinese (%)	84 (51.2)	34 (52.3)	2 (6.7)	**<0.001**
Number of Female (%)	108 (64.0)	39 (60.0)	19 (63.3)	0.70
BMI	22.5	22.3	22.8	0.79
Sweet Food Liking	5.3	4.9	5.1	0.20
Sour Food Liking	4.6	4.8	5.2	0.05
W	1.07a	1.03a	4.77b	**<0.001**
C	7.29b	4.06a	6.80b	**<0.001**
T	7.27c	4.18a	6.10b	**<0.001**
S	1.60b	1.11a	1.90b	**<0.01**
SC	4.88a	4.77a	4.33a	0.43
ST	4.87b	3.97a	5.30b	**<0.01**

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
