# Peer review of "Taste Interactions between Sweetness of Sucrose and Sourness of Citric and Tartaric Acid among Chinese and Danish Consumers"

_foods, 2020, doi:10.3390/foods9101425_

Round 1
Reviewer 1 Report
This is a very interesting, and soundly executed and analysed, piece of sensory research. The key issue I have is that there is no reference to whether any research ethics clearance obtained for the study.
Please clarify the ethics clearance status fro this research.
The quality of writing is in the main, excellent but there are a very few corrections that need to be addressed.
Abstract: Line 23 - put a capital C on Culture.
Results:
Line 450-451 - Rephrase as it is not reading clearly, it seems to be missing a verb, perhaps:
'Danish consumers, generally on average gave higher sweetness intensity rating than Chinese consumers did'.
Lines 479-481: The second clause of this sentence is not clearly worded and needs to be rephrased:perhaps,
Thus, Sweet Cluster 1 may be said to consist of consumers whom have a lower
sweetness sensitivity, why as consumers in this cluster respond less to sweetness, and experience a higher degree of sweetness suppression from acids.
Reviewer 2 Report
Materials and Methods
Some points are proposed for modification (shaded in yellow).
Line 179 - …of questions using a 7 point likert Likert scale about general sweet and sour food liking ("I like to eat sweet…
Line 183 - …Body Mass Index (BMI) was calculated based on consumer height and weight weight (kg)/(height…
Line 186 - …All data analysis analyses were performed using R [33] and RStudio [34]. Data formatting and plotting was…
Line 193 - …calculated using R package SensoMineR [37] and Tukeys Tukey's Honest Significant Differences (HSD)…
Line 195 - …effects and consumer effects as random effects. HSD were was determined using the R package lmerTest…
Line 207 - …are presented visually in Figure 2, using the ggplot2 package in R [42]. They are shown as differences…
Line 209 - …pairwise comparison of estimated marginal means (,) and p-values shown are adjusted using Bonferroni…
Line 216 - …interaction between these on either sweetness- or sourness ratings were was analysed using Mixed model…
Line 218 - … effects and the consumer effects were considered random effects. To further investigate the three-way…
Line 224 - …distances and Wards method using the R package cluster [46]. The nNumber of clusters were determined…
Line 230 - … Mean calculation and HSD was were performed similarly to the Overall Sample Evaluation Analysis. ANOVA…
Results and Discussion - I don't agree with Results and further Discussion thinking that's better combine Results and Discussion.
Line 237 - …sourness of samples are is shown in Table 3. For sweetness ratings, the sample only containing sucrose…
Line 241 - 242 - …acid. The sample containing only tartaric acid (T) were was placed between the two (,) and were not different from neither W nor C.
Line 245 - …as well as an acid (SC and ST). As expected, there is no difference between W and S in terms of sourness…
Legend of figure 1
(A) shows sourness intensity rating, and (B) shows a sweetness intensity rating.
Figure 1A shows that as
Line 262 - …expected, citric- and tartaric acid differs from the No acid condition, but are not different from each…
Line 266 - In Figure 1B, as expected, the No acid condition differed from the acids when sucrose were was present,…
Line 268 - …intensity, the acids gave a similar response, indicating a similar suppression effect from both citric- and…
Line 270 - …Figure 1B shows that the effect of the acids on sweetness intensity ratings are much larger when sucrose…
Line 299 - … differently. This was also indicated from in Figure 2.
Line 320 - …any differences, as the confidence intervals, overlap. There might be a trend towards the opposite…
Line 329 - …The cluster analyses are shown below (,) and reveals patterns of individual difference in ratings…
Line 334 - …the differences in sweetness ratings. Three clusters were found and PCA were was performed to further…
Line 369 - …for the non-sucrose containing samples. For the samples containing sucrose, the sweetness ratings is are…
Conclusions
Line 517 - …Further, this study investigated the perception of taste interactions between sweetness and sourness…
Line 519 - …of sweetness on sourness was found, but not of sourness on sweetness. A Further analysis revealed…
Line 522 - …Lastly, a cluster analysis showed differences in taste interactions for both sweetness and sourness…
Line 525 - …beverages are rather due to the individual than cultural differences. Thus, with a focus in the beverage…
Line 526 - …industry to the diversification of the product range, it might be fruitful to investigate not only different…
